# Epidemiology and Scenario Simulations of the Middle East Respiratory Syndrome Corona Virus (MERS-CoV) Disease Spread and Control for Dromedary Camels in United Arab Emirates (UAE)

**DOI:** 10.3390/ani14030362

**Published:** 2024-01-23

**Authors:** Magdi Mohamed Ali, Eihab Fathelrahman, Adil I. El Awad, Yassir M. Eltahir, Raeda Osman, Youssef El-Khatib, Rami H. AlRifai, Mohamed El Sadig, Abdelmalik Ibrahim Khalafalla, Aaron Reeves

**Affiliations:** 1UAE Ministry of Climate Change and Environment, Dubai 1509, United Arab Emirates; mmali@moccae.gov.ae; 2Department of Integrative Agriculture, College of Agriculture and Veterinary Medicine, United Arab Emirates University (UAEU), Al Ain 1551, United Arab Emirates; a.alawad@uaeu.ac.ae (A.I.E.A.); raeda_mohammed@uaeu.ac.ae (R.O.); 3Abu Dhabi Agricultural and Food Safety Authority ADAFSA, Abu Dhabi 52150, United Arab Emirates; yassir.eltahir@adafsa.gov.ae (Y.M.E.); abdelmalik.khalafalla@adafsa.gov.ae (A.I.K.); 4Department of Mathematical Sciences, College of Science, United Arab Emirates University (UAEU), Al Ain 1551, United Arab Emirates; youssef_elkhatib@uaeu.ac.ae; 5Institute of Public Health, College of Medicine, and Health Sciences (UAEU), Al Ain 1551, United Arab Emirates; rrifai@uaeu.ac.ae (R.H.A.); msadig@uaeu.ac.ae (M.E.S.); 6Center for Public Health Surveillance and Technology, RTI International, Research Triangle Park, Raleigh, NC 27709, USA; areeves@rti.org

**Keywords:** Middle East Respiratory Syndrome Corona Virus (MERS-CoV), disease spread, control strategy, dromedary camels, vaccination, costs

## Abstract

**Simple Summary:**

Middle East Respiratory Syndrome (MERS-CoV) is a coronavirus-caused viral respiratory disease. MERS-CoV is a zoonotic virus that spreads between animals and humans. The objectives of this research include simulation of MERS-CoV spread using a customized animal disease spread model (i.e., customized stochastic model for the United Arab Emirates (UAE); analyzing the MERS-CoV spread and prevalence based on camel age groups and identifying the control MERS-CoV strategies to aid the decision-maker in the selection of the optimum strategy to control the spread of the disease. The results of this research conclude that movement control is the optimum “best” strategy to control the spread of MERS-CoV.

**Abstract:**

Middle East Respiratory Syndrome (MERS-CoV) is a coronavirus-caused viral respiratory infection initially detected in Saudi Arabia in 2012. In UAE, high seroprevalence (97.1) of MERS-CoV in camels was reported in several Emirate of Abu Dhabi studies, including camels in zoos, public escorts, and slaughterhouses. The objectives of this research include simulation of MERS-CoV spread using a customized animal disease spread model (i.e., customized stochastic model for the UAE; analyzing the MERS-CoV spread and prevalence based on camels age groups and identifying the optimum control MERS-CoV strategy. This study found that controlling animal mobility is the best management technique for minimizing epidemic length and the number of affected farms. This study also found that disease dissemination differs amongst camels of three ages: camel kids under the age of one, young camels aged one to four, and adult camels aged four and up; because of their immunological state, kids, as well as adults, had greater infection rates. To save immunization costs, it is advised that certain age groups be targeted and that intense ad hoc unexpected vaccinations be avoided. According to the study, choosing the best technique must consider both efficacy and cost.

## 1. Background

Middle East Respiratory Syndrome (MERS-CoV) is a coronavirus-caused viral respiratory disease. While it was initially identified in 2012 in Saudi Arabia, primarily affecting camels and humans in the Middle East, it has since been reported in other parts of the world because of travel-related cases. It is important to note that MERS primarily spreads through close contact with infected individuals or camels or their respiratory secretions, such as coughing or sneezing. The virus is less contagious than other respiratory viruses like the common cold or influenza. Human MERS symptoms can range from moderate to severe, including fever, cough, shortness of breath, and pneumonia. In some cases, it can result in complications such as kidney failure or even death, particularly among individuals with pre-existing medical issues or compromised immune systems (Immunocompromised) [1].

Since July 2013, when the UAE reported the first case of MERS-CoV, 94 confirmed cases (including this new case) and 12 deaths have been reported. Globally, the total number of laboratory-confirmed MERS-CoV cases reported to WHO since 2012 is 2605, including 936 associated deaths, with a case-fatality ratio of 36% as of July 2023. In total, cases were reported from 27 countries globally. Most of the reported cases have occurred in countries in the Arabian Peninsula [2]. The bar chart in Figure 1 shows the number of MERS-COV infections and fatalities from April 2012 to July 2019. As a result of recurrent outbreaks and control measures, the number of cases and deaths fluctuates between rising and falling. The overall number of cases has dropped as infection management in hospital settings has improved, and a history of animal interaction is being recorded more consistently in case investigations. As shown in Figure 2, the proportion of primary cases involving animal exposure appeared to rise until June 2022. However, this proportion shifted in July 2022 when indirect exposure increased [3].

MERS-CoV is a zoonotic infection in which bats and dromedary camels play essential roles in its emergence and epidemiology. Camel-to-human MERS-CoV transmission has been documented. However, such documentation is, in general, inefficient. The precise transmission process, as is the involvement of additional intermediate hosts [4], is unknown. Human-to-human transmission is conceivable and has occurred primarily among close contacts and in healthcare settings. Camels are an important reservoir for MERS-CoV persistence and a critical source of human MERS infection [5,6]. As indicated in Figure 3, There is a scarcity of evidence characterizing the MERS-CoV transmission cycle in diverse hosts [7]. Well-designed large-scale studies are needed to define the transmission chain of MERS-CoV [8]. Understanding the transmission spatial distribution will help decision makers formulate the proper policy to prevent the transmission of MERS-CoV across country borders. 

The camel population in the United Arab Emirates (UAE) increased from 358,027 to more than 461,788 during the last two decades, from 2010 to 2019, based on FAO statistics, distributed across the country (Figure 4 and Figure 5) [9]. Camels have been at the very center of life in the UAE for decades, playing important roles in transportation, cuisine, and entertainment, among other things. As such, they symbolize Emirati heritage and remain integral to the country’s customs and cultural practices (for example, camel racing). In recent years, camels in the UAE have become part of food production and food security regarding camels’ milk and meat production. The Food and Agriculture Organization (FAO), 2023, showed that the camel population in the UAE in 2021 exceeded half a million heads. Camels producing raw and processed milk in the country are estimated to be about 350 thousand heads. Camels produced for camels’ meat is estimated at 217 thousand heads. Camels’ meat and milk production is expanding in the country, so considering the biosecurity of such a sector is highly important [10].

In UAE, a high seroprevalence (97.1) of MERS-CoV in camels was reported [11]. Studies performed in the Emirate of Abu Dhabi included camels in zoos, public escorts, slaughterhouses, and the borders with Saudi Arabia and Oman, which showed an overall MERS-CoV—PCR positive rate of 1.6% [12]. The UAE ranks third in proven human cases globally, After Saudi Arabia and South Korea [13]. Risk factors for MERS-CoV seropositivity among animal market and slaughterhouse workers in Abu Dhabi included working as a camel salesman, handling live camels or their waste, and having diabetes [14]. Despite the report of MERS-CoV in both humans and camels in UAE, no simulation spread studies were conducted to explore suitable strategies to control MERS-COV.

## 2. Research Objectives

The primary goal of this study is to simulate MERS-CoV spread using a customized animal disease spread model (i.e., a customized stochastic model for the UAE) and to analyze MERS-CoV spread and prevalence based on camel age groups, as well as to identify control MERS-CoV strategies implications and to estimate the relevant government costs (costs of such strategies control). Accruing knowledge about the presence of MERS-CoV in the animal reservoir is a crucial first step to developing intervention and control measures to prevent human infections.

## 3. Data and Methods

This section contains a detailed description of the study data and the method used in this research.

### 3.1. Data 

The Animal Health Division of the Abu Dhabi Agriculture and Food Safety Authority (ADAFSA) provided data on the livestock population. Data on animals were collected from 24,836 holdings and farms. All species in this study had a combined population of over 3 million heads. A cluster of animals called a “unit” is the basis of simulation in the suggested stochastic model of the disease. A product type, number of animals, point location (given in terms of longitude and latitude), and disease state are all part of a unit. Production type is a collection of herds with similar disease transmission probabilities, disease manifestation, disease detection probabilities, and control strategies. Production type is a group of animal species and management practices applied to each herd.

### 3.2. Methodology and Simulation Scenarios

Epidemiologic modeling is a typical approach to simulating and constructing different scenarios to predict the potential impact of contagious disease outbreaks, such as MERS-CoV, in domesticated animal populations. The information generated by these scenarios is essential, and policymakers can use it to control diseases and prepare for early and ongoing disease management and eradication efforts. Several spatially explicit stochastic epidemic simulation models have been created to assess the transmission of highly contagious animal diseases and model disease outbreaks [15,16]. 

The North American Animal Disease Spread Model (NAADSM) was the framework and software used in this study, which was adapted and customized. The software was created to simulate the spread and control of foreign animal diseases in a herd of vulnerable livestock [17]. Calves showed a significantly greater viral RNA frequency and a higher virus isolation rate than subadults (2–4 years of age) (2, p 0.05). Calves have a greater rate of viral isolation, indicating that they are more infectious [18,19]. Accordingly, the following categories of units (herds) entered into the modified NAADSM:Camel kids less than one year old;Young camels at age 1 to less than four years old;Adult camels four years and older.

Three alternative simulation scenarios (1, 2, and 3) were run using the NAADSM as follows to examine the effects of control techniques for rapidly eliminating MERS-CoV from the UAE (in terms of lowering spread as well as anticipated direct government cost):This scenario demonstrates the condition of no control measures; it represents a strategy that recognizes the initial situations and serves as a foundation for the other strategies.This scenario depicts an instance in which efforts and control strategies are based on no or limited animal movement.The third scenario represents a strategy based on intensive vaccination and strict animal movement controls.


**
*Disease states*
**


This study used the NAADSM’s seven unique illness states, as depicted in Figure 6. When a susceptible unit in the model becomes infected, the disease will become latent until a disease control intervention is adopted. Furthermore, unless disease management measures are implemented, an infected unit will develop naturally from a dormant state to a sub-clinically infectious one. Disease progression will follow the natural progression, as indicated in the outer loop (see image below). Still, disease control implementation may disrupt the natural disease cycle, as shown inside the loop. A period spent in a particular disease state for each herd is required to run this model [20].


**
*Disease Spread*
**


Disease spread can occur by direct, indirect, or airborne contact.

***Direct contact spread parameters:*** (Animal movement or shipment between units)

Animal shipment mean rate (number of recipient units per source unit per day).Distance traveled (probability density function per kilometer).Delay in shipping (probability density function per day).The probability of infection of the receiving unit due to exposure to an infected unit.Movement rate multiplier (a scalar value based on the number of days since the outbreak was first detected).


**
*Indirect Contact Spread Parameters:*
**


The movement of individuals, supplies, equipment, automobiles, livestock products, and so on among units is replicated in the same manner as direct contact, except that only sub-clinically infected and clinically infectious units can operate as the source of infection, not latent units. The parameters for indirect contact are comparable to those for direct contact, although they differ. 


**
*Detection and Reporting:*
**


Detection refers to identifying and reporting infected herds based on the appearance of clinical signs. Two probabilities affect the overall chance that an infected herd will be detected:The probability of observing clinical signs in a herd, given the number of days a herd has exhibited clinical signs.The probability that the owner or veterinarian will report the disease to the animal health authorities given the number of days since the disease was first detected and reported anywhere in the population.


**
*Tracing Out:*
**


NAADSM simulates trace-out investigations when an infected unit is found by tracing one level forward.

### 3.3. Control Measures/Strategies

Stochastic Model: The NAADSM model uses these measures to control the disease: quarantine, destruction, and vaccination.


**
*Quarantine:*
**


Units are confined in the model for one or more reasons: an infectious unit is quarantined immediately on the following detection day, and units traced out are similarly quarantined.


**
*Vaccination Parameters:*
**



The number of units that must be discovered before vaccination may begin (the number of units that have been detected).Vaccination capacity (relational function: number of units that can be vaccinated as a function of the number of days after the epidemic was first detected).Vaccination priorities (order of importance for unit vaccination).The radius of the vaccination ring, whether or not the ring is activated, and other considerations.



**
*Vaccination Costs parameters:*
**


Even though there is currently no developed vaccine against MERS-CoV, in this study, vaccination is considered a control strategy if a vaccine becomes available. The model can quantify direct costs related to destruction and immunization to compare the costs of various control techniques. Here are some cost input factors that NAADSM uses to calculate direct government expenditures related to vaccination. Parameters associated with vaccination include the following:The number of animals that can be vaccinated at the baseline cost.The cost of vaccination at the baseline cost.When the number of animals vaccinated exceeds the threshold, additional expenditures are paid.The cost of establishing a vaccination location.

See Appendix A: prodcution type, disease, contract spread, detection, tracing, exam for clinical signs, zones, vaccination, cost accounts sheets.

## 4. Results and Discussion

Middle East respiratory syndrome CoV (MERS-CoV) emerged in 2012 as a source of severe respiratory illnesses with high mortality rates in humans. So far, several investigations on the virus’s prevalence and characteristics have been undertaken in the United Arab Emirates. In 2015, A molecular analysis of MERS-CoV in the UAE revealed the virus’s prevalence in dromedary camels. The researchers employed a real-time PCR test to screen nasal swab samples and found a 1.6% detection rate. The high incidence of MERS-CoV antibodies may protect dromedary camels. Nevertheless, it is unknown if these antibodies are effective. The absence of clinical indications or observed mortalities in MERS-CoV-infected camels could point to camels acting as reservoirs for human infection. The virus was found in camels from slaughterhouses in Abu Dhabi Emirate and along the borders with Saudi Arabia and Oman. A study by Yusof et al. implied transmission of MERS-CoV in humans and camels due to epidemiologic links between human disease and camel-borne pathogens [13]. In another study by Killerby et al., MERS-CoV was found in nasal and rectal swabs from infected camels for up to two weeks following discovery. The virus, however, was not found in milk or water samples from infected farms. The researchers determined that MERS-CoV was not highly transmissible from dromedaries to people and that camels may play a role in human viral transmission [6].

Research in Abu Dhabi, UAE, examined 376 camels for MERS-CoV. In week one, 109 positive camels were found, and 139 samples yielded 126 whole or almost full genomes. Evidence suggests that MERS-CoV infection in humans is caused by the continual introduction of diverse camel lineages [12]. Similarly, Li, Y. et al. performed metagenomic sequencing on nasopharyngeal swab samples from 108 MERS CoV-positive dromedary camels in Abu Dhabi, UAE. There were 846.7 million high-quality reads collected, with 0.34% relating to viral sequences. Sequences were found in 13 taxa and 10 viral families, identifying five potentially new virus species or strains. At least two recently reported camel coronaviruses co-infected 92.6% of the camels [21].

Previous research also documented the virus’s prevalence in various age groups in the UAE. For instance, Wernery et al. reported that the condition is an acute, epidemic, and time-limited infection in calves four years of age and older in the camel population in the UAE. The study comprised around 800 dromedaries of various ages and 15 mother–calf couples. Because animal ages ranged between sites, the samples were classified based on the ages of the camels rather than the sampling site. MERS-CoV antibodies were found in 96% of dromedaries over two years old and 80% of calves. Reverse transcription PCR (RT-PCR) testing found that calves had a greater incidence of MERS-CoV RNA, and virus isolation was only successful in camels under four. The greater prevalence of virus isolation shows that calves are more infectious [18]. Similarly, another study in UAE found that MERS-CoV seropositivity in dromedary calves increases with age, reaching nearly 100% in adult dromedaries [19]. This is consistent with the finding that MERS-CoV infections only develop in dromedary calves after their mothers’ antibodies have vanished [11]. 

In this study, simulation results for scenario 1 show the absence of control measures, scenario 2 shows the efforts and control strategies focused on no or limited animal movement, and scenario 3 shows a vaccination method based on strict animal movement controls, as shown in Figure 7 and Figure 8 and Table 1 below. The results showed that the outbreak durations for scenarios 1, 2, and 3 are 228, 36, and 35 days, respectively.

The total number of susceptible animals was found to be 157.5 thousand animals across the three scenarios: 1, 2, and 3. In contrast, the total number of animals that became latent and naturally immune and exhibited clinical and subclinical symptoms was reported to be 117 thousand 434 433 animals throughout an iteration for the three scenarios 1, 2, and 3, respectively. Additionally, the number of animals initially infected at the beginning of an iteration was reported to be 200 animals across the three scenarios. The total number of animals in all units infected throughout an iteration by direct contact was reported to be 116.9 thousand, 234, and 233 animals for the three scenarios 1, 2, and 3, respectively. 

Tracing was also addressed as the number of animals exposed to an infected herd and successfully traced forward and back throughout an iteration. Accordingly, the total number of animals in all units successfully traced after direct contact was reported to be 74 thousand, 256, and 255 animals in scenarios 1, 2, and 3, respectively. On the other hand, the total number of animals successfully identified by tracing after indirect contact was 228.9 thousand, 213, and 205 animals in scenarios 1, 2, and 3, respectively. Moreover, the total number of animals in all units detected by clinical signs throughout an iteration was 117 thousand 434, 433 animals in scenarios 1, 2, and 3, respectively. In contrast, the total number of animals detected by diagnostic testing was found to be 67 thousand in scenario 1, and 178 and 157 animals in scenarios 2 and 3. 

The simulation scenarios detailed the control strategies that can be applied against the Middle East Respiratory Syndrome Corona Virus (MERS-CoV), and this study discusses the outcomes and compares the scenarios’ outcomes, as illustrated in Table 1 below. These results show that the control strategy that restricted animal movement (scenario 2) compared to no control measure (scenario 1) reduced the outbreak duration on average of the model’s 1000 iterations from 228 days to 36 days, or by 88%. The number of infected animals reduced from 117 thousand to only 234 animals. Meanwhile, in the control strategy (scenario 3), which applies vaccination and animal movement controls, the number of animals infected reduced from 117 thousand animals to only 233 animals in scenario 3 compared to scenario 1. The total number of animals directly exposed to an infected herd over the iterations was also reduced from 74 thousand to 234, whereas the indirectly exposed animals reduced from 228.9 thousand to only 213 animals. This indicates that reliance on no-movement measures alone would contribute enough to achieve MERS-CoV eradication.

The comparison of Scenarios 2 and 3 highlights the differences caused by the introduction of extensive vaccination as a disease control approach. Due to vaccination, out of 4536 vaccinated animals, 2857 animals became vaccine-immune, and the outbreak duration reduced from 36 to 35 days, or by 3%. The number of animals infected with direct or indirect contact over the model’s 1000 iterations reduced by 0.43%. In addition, the total number of animals directly exposed and successfully traced forward and back of an iteration reduced from 256 in scenario 2 to 255 in scenario 3, whereas indirectly exposed animals reduced by 4%. As illustrated in Figure 8, the epic curves of scenarios 2 and 3 exhibit a sharp increase to the peak and, subsequently, a slower fall of newly detected animals; in addition, the epic curve in scenario 2 maintained a flat and prolonged curve throughout the outbreak. From the comparison of the two scenarios, all other indicators about disease transmission and the effect of the vaccination approach revealed changes that ranged from 0% (no change) to −4%. This suggests that a dependence on either movement restriction or vaccination and movement restriction measures would be sufficient to stop the spread of MERS-CoV.

A systematic review and meta-analysis of the Prevalence of Middle East Respiratory Syndrome Coronavirus in dromedary camels and small ruminants in the Gulf Cooperation Council countries and Yemen and Iraq was conducted and reported in PRISMA at the University of York Prospero system. The study was cross-sectionally designed and conducted in PubMed, Embase, Scopus, Web of Science, and Cochrane from their inception to September 2021. The weighted prevalence and its 95% confidence intervals of MERS-CoV virus structured around the crude prevalence, species, and country were quantified. According to Table 2 below, the adult animal category has an average Sero Prevalence of 93% of the Middle East Respiratory Syndrome Coronavirus, followed by the young and kid categories with 87% and 70%, respectively.

Accordingly, this study simulated and analyzed the disease spread and identified the control strategies outcomes based on the age of the camel’s disease spread and the prevalence parameters obtained from the systematic review. The simulation results are illustrated in Figure 9 and in Table 3, Table 4 and Table 5. The total number of animals that become susceptible over the iterations was reported to be 121.8, 21.6, and 14 thousand animals for the kids, young, and adult camel categories, respectively. The number of camel kids that were infected reduced sharply from 9 thousand in scenario 1 to only 16 and 15 animals in scenarios 2 and 3, respectively, while for the young camels, it reduced from 16.5 thousand animals in scenario 1 to 36 animals in scenario 2 and 35 animals in scenario 3. Similarly, the infected adult camels reduced from 91.5 thousand to only 182 and 183 animals in the three scenarios 1, 2, and 3, respectively. 

As a result of moving from the restricted or no movement control strategy (scenario 2) to intensive vaccination and no movement control strategies (scenario 3), the number of animals that become latent and that showed clinical and subclinical signs over the iterations reduced by 0.5%, 3%, and 1% for the Kids, young, and adult camels, respectively. Similarly, the number of camel kids exposed directly to an infected herd over the model’s 1000 iterations and identified by tracing reduced by 9%, while the young camels reduced by 3%, and adult camel categories reduced by 5%. The indirectly exposed camel kids identified by tracing decreased by 2%, while the young camels reduced by 4%, and adult camel categories reduced by 1%. Additionally, the total number of animals detected by clinical signs showed a reduction of 0.5%, 3%, and 1% for the three kid, young, and adult camel categories, while the camel kids detected by diagnostic testing reduced by 8%.

As demonstrated in Table 6, the total number of vaccine-immune animals was higher than that of animals that become naturally immune by 4 and 14 times for the young and adult camel categories, respectively. In contrast, camel kids become more naturally immune than vaccine immune. Moreover, the total number of vaccine-immune animals was reported to be 35%, 23%, and 73% for the kid, young, and adult camel categories. This result indicates that the adult camel category, followed by the camel kids category, responded more to the intensive vaccination and limited movement control strategy.

The costs related to vaccination are displayed in Table 7 below. The findings of the scenario simulation revealed that, while the average cost of vaccination is 44,677 USD, the average fixed cost of setting up a vaccination site for each vaccinated unit is 141,325 USD—the total number of animals to be vaccinated determined by the cost of vaccination for each animal. Only a base vaccination cost is charged for each animal up to a certain threshold. An extra fee is charged for each additional animal. As a result, the total cost of vaccination was projected to be 186,002 USD on average of the iterations, around 2 million USD being the highest cost of all iterations of scenario 3.

Further studies are needed to investigate the origin of MER-CoV, not only the host of the virus (the camels). The origins of the virus may include birds and ticks. We are studying the ecological connection and environmental conditions that may increase or reduce the spread of the virus. Furthermore, studies are needed to present the importance of the disease and the appropriate scientific methods, clinical trials, and protocols to develop a vaccine against MERS CoV for both animals and humans.

## 5. Conclusions

Comparing the proposed control strategy, this study found that scenario 2 (animals’ movement control) is the optimum strategy compared to the other simulated scenarios. In such a scenario, the outbreak duration was reduced from 288 days in scenario 1 to only 36 days in the optimum scenario 2. Due to the movement control strategy, infected farms were reduced from 3141 farms or 116,982 camels to 6 farms or 234 camels. Scenario 3 is not considered an optimum scenario because, despite the high cost of vaccination, the control policy/measure did not include a significant reduction in the number of infected animals. 

The results also showed a large variation in the disease spread and its implications/consequences of the diseases spread between camels in three age categories: camel kids that are less than one-year-old, young camels aged one to less than four years old, and adult camels that are four years and older. Both the kids and adults showed higher infection levels than young ones (between 1 and 4 years old). This is primarily due to the level of immunity in these two categories, as the camel kids usually do not reach the full development of their immune system. In contrast, the older ones usually tend to be immunocompromised. With this finding, it is recommended to target these two age categories by giving them higher priority when applying movement control and vaccination. Furthermore, this finding indicates that to minimize the cost of vaccination, it is recommended to not apply intensive ad hoc unplanned vaccination. Instead, the effective vaccination campaign may test for naturally immune animals, exclude those from vaccination, and apply vaccination only to kids and adult camels. This study showed that selecting the optimum strategy requires considering both the strategy’s effectiveness and its cost through a comparison of the outcome of the strategy compared to its cost. Animal disease spread simulation can aid policymakers in formulating effective eradication and increasing biosecurity.

## Figures and Tables

**Figure 1 animals-14-00362-f001:**
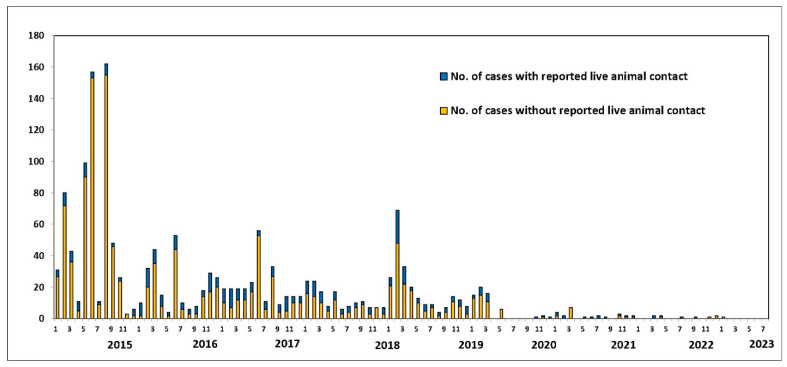
Human MERS-CoV cases worldwide, broken down by potential source of exposure (in percent). Source: Food and Agriculture Organization (FAO). Retrieved from https://www.fao.org/animal-health/situation-updates/mers-coronavirus, accessed on 15 July 2023 [3].

**Figure 2 animals-14-00362-f002:**
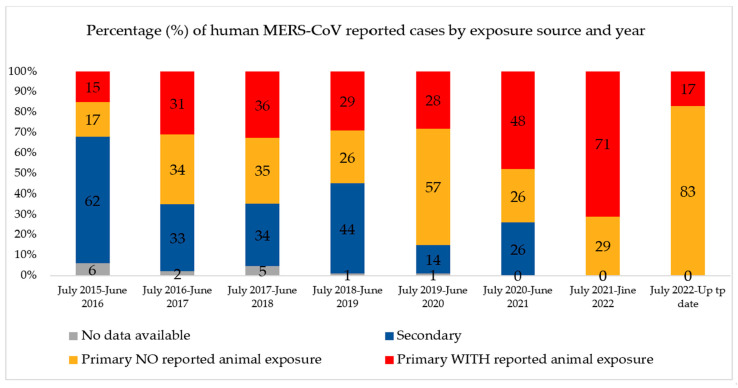
Percentage of human MERS-CoV reported cases by exposure source and years globally. Source: Food and Agriculture Organization (FAO). Retrieved from https://www.fao.org/animal-health/situation-updates/mers-coronavirus, accessed on 15 July 2023 [3].

**Figure 3 animals-14-00362-f003:**
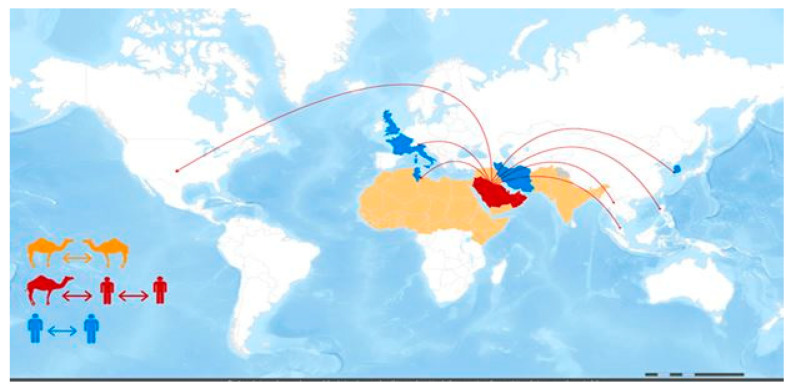
*MERS-CoV transmission and geographical range*. Reprinted from emergency preparedness, response—Middle East respiratory syndrome-coronavirus (MERS-CoV), WHO, MERS-CoV transmission, and geographical range. Website https://www.fao.org/animal-health/situation-updates/mers-coronavirus, accessed on 12 July 2023 [7].

**Figure 4 animals-14-00362-f004:**
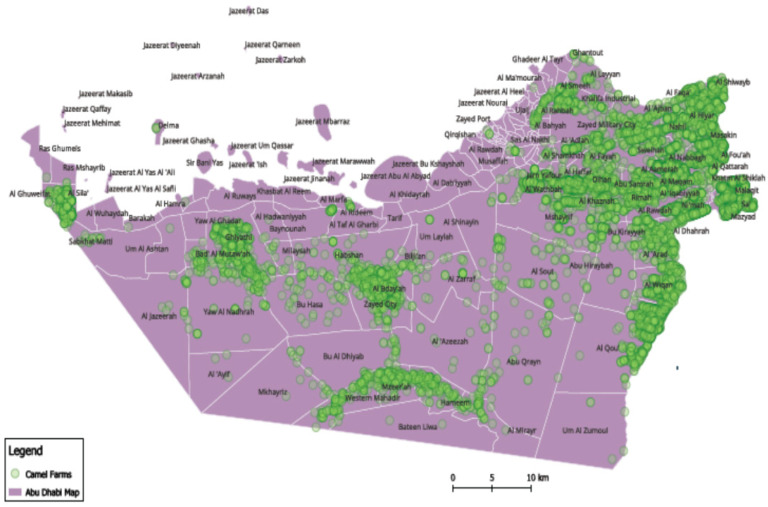
Camel spatial distribution in the United Arab Emirates (UAE). Note: Map developed by the authors.

**Figure 5 animals-14-00362-f005:**
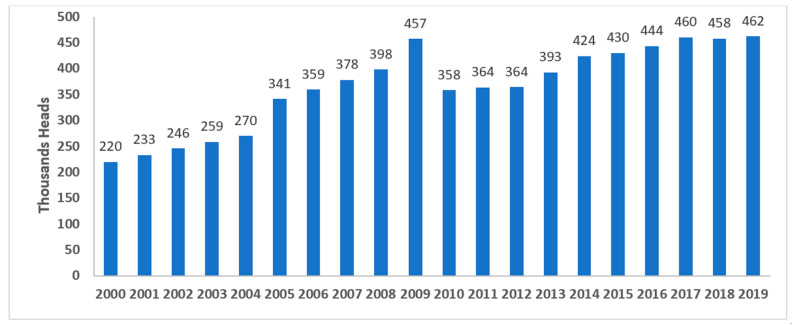
Camels’ population in the United Arab Emirates (UAE) in 2019, according to Food and Agriculture Organization (FAO) statistics [9]. **Note:** Graph developed by the authors.

**Figure 6 animals-14-00362-f006:**
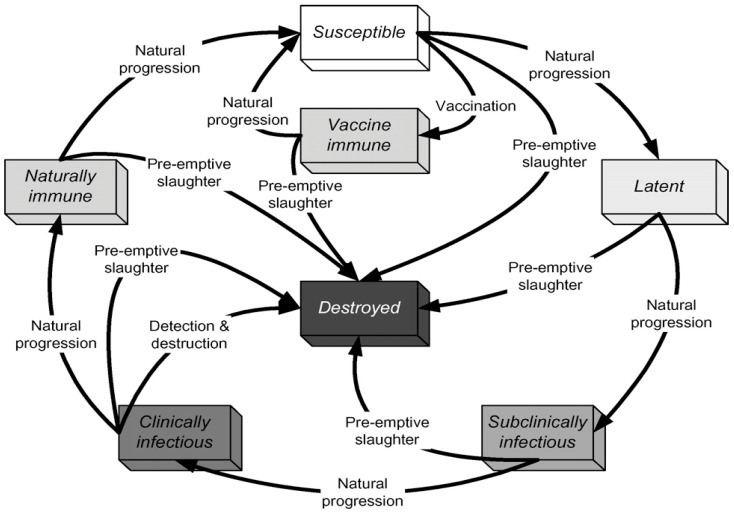
States and transitions simulated by Customized United Arab Emirates AE NAADSM. Source: The North American Animal Disease Spread Model NAADSM. Available online: https://pubmed.ncbi.nlm.nih.gov/17614148/ (accessed on 15 February 2023) [20].

**Figure 7 animals-14-00362-f007:**
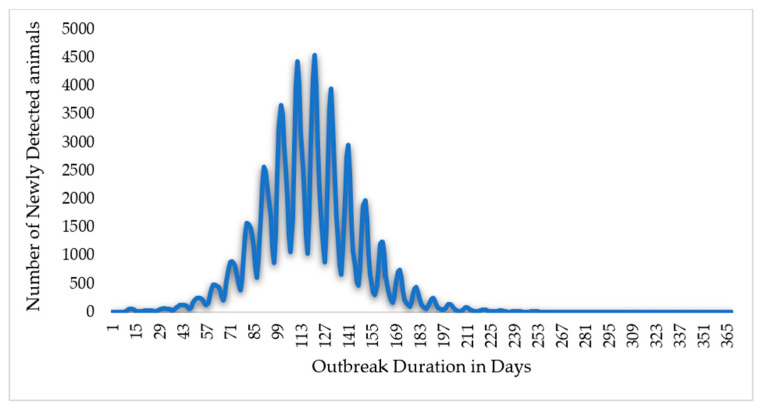
The Epi curve output for scenario 1 (without control measures). In this scenario, the total outbreak duration in days is 365 days, peaking at 121 days.

**Figure 8 animals-14-00362-f008:**
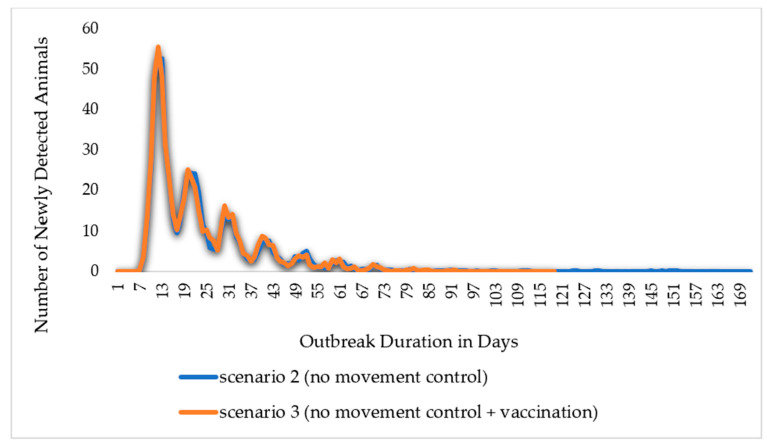
The Epi curve output for the scenario with control measures. The outbreak duration in scenario 2 (no movement control) reached 171 days, and in scenario 3 (no movement control + vaccination), it reached 118 days.

**Figure 9 animals-14-00362-f009:**
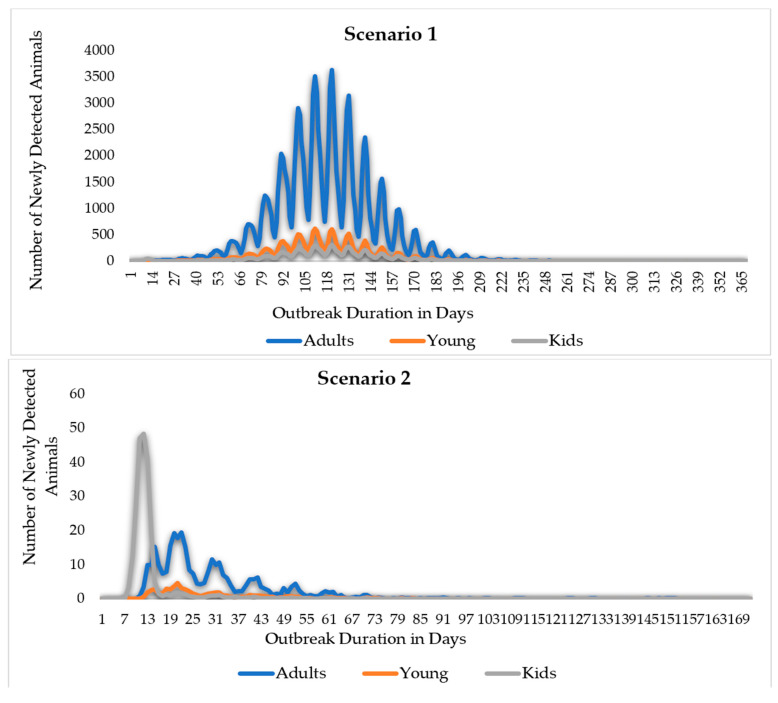
Middle East Respiratory Syndrome (MERS) scenarios, EPI curves by scenario and age. The adult camel’s curve is symmetric in scenario 1, while it is more skewed to the left in scenarios 2 and 3. After the 15th day of the outbreak duration, the EPI curve of the camel kids’ categories drops dramatically in scenarios 2 and 3.

**Table 1 animals-14-00362-t001:** Simulation scenarios’ outcomes/results for all camels.

Output Description	Scenario 1		Scenario 2		Scenario 3	
	(No Control)		(No Movements)		(No Movements + Vaccination)	
	Average Number of Farms	Average Number of Camels	Average Number of Farms	Average Number of Camels	Average Number of Farms	Average Number of Camels
Total number that becomes susceptible	4442	157,539	4442	157,539	4442	157,539
Total number that becomes latent	3142 (788)	117,182 (29,375)	7 (9)	434 (332)	7 (8)	433 (397)
Total number that becomes subclinical	3142 (788)	117,182 (29,375)	7 (9)	434 (332)	7 (8)	433 (397)
Total number that becomes clinical	3142 (788)	117,182 (29,375)	7 (9)	434 (332)	7 (8)	433 (397)
Total number that becomes naturally immune	3142 (788)	117,182 (29,375)	7 (9)	434 (332)	7 (8)	433 (397)
Total number that becomes vaccine-immune	0	0	0	0	37 (66)	2857 (2714)
Total number initially infected	1(0)	200 (0)	1 (0)	200 (0)	1 (0)	200 (0)
Total number that become infected over the course (not including initial infection)	3141 (788)	116,982 (29,375)	6 (9)	234 (332)	6 (7)	233 (397)
Total number identified by tracing of direct contact	2270 (572)	74,241 (18,833)	6 (9)	256 (345)	7 (8)	255 (378)
The total number identified by tracing indirect contact	5959 (1479)	228,895 (57,691)	5 (6)	213 (282)	4 (6)	205 (318)
Total number detected by clinical signs	3142 (788)	117,182 (29,375)	7 (9)	434 (332)	7 (8)	433 (397)
Total number detected by diagnostic testing	1831 (460)	67,057 (16,972)	5 (6)	178 (243)	4 (6)	178 (273)
Total number vaccinated for any reason	0	0	0	0	72 (82)	4536 (2840)
Duration of the outbreak in the specific iteration	228 (56)	228 (56)	36 (15)	36 (15)	35 (14)	35 (14)

Mean (SD). Source: Study simulations results.

**Table 2 animals-14-00362-t002:** Camel’s population and prevalence parameters were extracted from a meta-analysis.

Parameter	Number of Observations	Mean	Median	Standard Deviation	Range	Coefficient of Variation (%)
Female PCR Prevalence %	7	10.3	4.9	10.7	30.0	103.7
Female Sero Prevalence %	8	86.0	84.1	10.7	33.5	12.5
Male PCR Prevalence %	5	15.2	17.6	11.3	25.1	74.4
Male Sero Prevalence %	7	72.9	83.5	30.2	86.0	41.4
Calf, juvenile\lamb\kid PCR Prevalence %	14	37.7	35.2	24.8	82.6	65.8
Calf, juvenile\lamb\kid Sero Prevalence %	11	70.2	84.0	25.8	85.8	36.8
Adults; 1–3 years PCR Prevalence %	3	3.0	2.9	0.4	0.8	13.7
Adults; 1–3 years Sero Prevalence %	3	87.0	86.5	9.3	18.6	10.7
Adults; 4 yrs. and above PCR Prevalence %	2	12.1	12.1	13.6	19.2	112.2
Adults; 4 yrs. and above Sero Prevalence %	4	93.1	95.4	7.9	18.3	8.5
Adults: age not specified PCR Prevalence %	11	13.0	9.9	10.6	31.4	81.6
Adults: age not specified Sero Prevalence %	9	72.5	83.8	37.3	92.0	51.5
PCR Prevalence %	27	26.9	21.2	25.6	96.9	95.3
Sero Prevalence %	19	64.3	85.8	42.0	99.3	65.3

Source: Unpublished meta-analysis results by the authors.

**Table 3 animals-14-00362-t003:** Camel kids’ simulation scenarios’ outcome results.

Output Description	Scenario 1		Scenario 2		Scenario 3	
	(No Control)		(No Movements)		(No Movements + Vaccination)	
	Average Number of Farms	Average Number of Camels	Average Number of Farms	Average Number of Camels	Average Number of Farms	Average Number of Camels
Total number that becomes susceptible	1480	14,172	1480	14,172	1480	14,172
Total number that becomes latent	924 (232)	9196 (2270)	3 (2)	216 (28)	2 (2)	215 (26)
Total number that becomes subclinical	924 (232)	9196 (2270)	3 (2)	216 (28)	2 (2)	215 (26)
Total number that becomes clinical	924 (232)	9196 (2270)	3 (2)	216 (28)	2 (2)	215 (26)
Total number that becomes naturally immune	924 (232)	9196 (2270)	3 (2)	216 (28)	2 (2)	215 (26)
Total number that becomes vaccine immune	0	0	0	0	16 (25)	145 (235)
Total number initially infected	1 (0)	200 (0)	1 (0)	200 (0)	1 (0)	200 (0)
Total number that becomes infected over the course (not including initial infection)	923 (232)	8996 (2270)	2 (2)	16 (28)	1 (2)	15 (26)
Total number identified by tracing of direct contact	732 (185)	7122 (1814)	2 (3)	64 (102)	2 (2)	58 (98)
The total number identified by tracing indirect contact	1476 (373)	14,780 (3761)	1 (2)	41 (83)	1 (2)	40 (86)
Total number detected by clinical signs	924 (232)	9196 (2270)	3 (2)	216 (28)	2 (2)	215 (26)
Total number detected by diagnostic testing	531 (134)	5165 (1320)	1 (2)	12 (23)	1 (2)	11 (28)
Total number vaccinated for any reason	0	0	0	0	24 (27)	411 (259)
Duration of the outbreak in the specific iteration	228 (56)	228 (56)	36 (15)	36 (15)	35 (14)	35 (14)

Source: Study simulations results.

**Table 4 animals-14-00362-t004:** Young camels simulation scenarios’ outcome results.

Output Description	Scenario 1		Scenario 2		Scenario 3	
	(No Control)		(No Movements)		(No Movements + Vaccination)	
	Average Number of Farms	Average Number of Camels	Average Number of Farms	Average Number of Camels	Average Number of Farms	Average Number of Camels
Total number that becomes susceptible	1481	21,550	1481	21,550	1481	21,550
Total number that becomes latent	1120 (281)	16,457 (4136)	2 (4)	36 (59)	2 (3)	35 (61)
Total number that becomes subclinical	1120 (281)	16,457 (4136)	2 (4)	36 (59)	2 (3)	35 (61)
Total number that becomes clinical	1120 (281)	16,457 (4136)	2 (4)	36 (59)	2 (3)	35 (61)
Total number that becomes naturally immune	1120 (281)	16,457 (4136)	2 (4)	36 (59)	2 (3)	35 (61)
Total number that becomes vaccine-immune	0	0	0	0	6 (19)	141 (338)
Total number initially infected	0	0	0	0	0	0
Total number that becomes infected over the course (not including initial infection)	1120 (281)	16,457 (4136)	2 (4)	36 (59)	2 (3)	35 (61)
Total number identified by tracing of direct contact	874 (221)	12,631 (3217)	2 (4)	37 (61)	2 (3)	36 (59)
The total number identified by tracing indirect contact	2385 (600)	35,714 (9028)	2 (2)	27 (41)	2 (2)	26 (45)
Total number detected by clinical signs	1120 (281)	16,457 (4136)	2 (4)	36 (59)	2 (3)	35 (61)
Total number detected by diagnostic testing	672 (170)	9828 (2501)	2 (3)	28 (43)	2 (3)	28 (46)
Total number vaccinated for any reason	0	0	0	0	24 (27)	618 (386)
Duration of the outbreak in the specific iteration	228 (56)	228 (56)	36 (15)	36 (15)	35 (14)	35 (14)

Source: Study simulations results.

**Table 5 animals-14-00362-t005:** Adults’ camels simulation scenarios’ outcomes results.

Output Description	Scenario 1		Scenario 2		Scenario 3	
	(No Control)		(No Movements)		(No Movements + Vaccination)	
	Average Number of Farms	Average Number of Camels	Average Number of Farms	Average Number of Camels	Average Number of Farms	Average Number of Camels
Total number that become susceptible	1481	121,817	1481	121,817	1481	121,817
Total number that becomes latent	1098 (276)	91,529 (23,003)	2 (3)	182 (269)	2 (3)	183 (337)
Total number that become subclinical	1098 (276)	91,529 (23,003)	2 (3)	182 (269)	2 (3)	183 (337)
Total number that become clinical	1098 (276)	91,529 (23,003)	2 (3)	182 (269)	2 (3)	183 (337)
Total number that becomes naturally immune	1098 (276)	91,529 (23,003)	2 (3)	182 (269)	2 (3)	183 (337)
Total number that becomes vaccine immune	0	0	0	0	16 (24)	2571 (2198)
Total number that are initially infected	0	0	0	0	0	0
Total number that become infected over the course (not including initial infection)	1098 (276)	91,529 (23,003)	2 (3)	182 (269)	2 (3)	183 (337)
Total number identified by tracing of direct contact	664 (168)	54,487 (13,904)	2 (3)	154 (241)	2 (2)	161 (284)
The total number identified by tracing indirect contact	2098 (527)	178,401(45,066)	2 (2)	146 (216)	2 (2)	138 (245)
Total number detected by clinical signs	1098 (276)	91,529 (23,003)	2 (3)	182 (269)	2 (3)	183 (337)
Total number detected by diagnostic testing	628 (158)	52,064 (13,248)	2 (2)	138 (204)	2 (2)	140 (237)
Total number vaccinated for any reason	0	0	0	0	24 (27)	3506 (2196)
Duration of the outbreak in the specific iteration	228 (56)	228 (56)	36 (15)	36 (15)	35 (14)	35 (14)

Source: Study simulations results.

**Table 6 animals-14-00362-t006:** Vaccinated and naturally immune camels across the three simulations.

	Scenario 1			Scenario 2			Scenario 3		
	Kids	Young	Adults	Kids	Young	Adults	Kids	Young	Adults
Naturally immune	9196	16,457	91,529	216	36	182	215	35	183
Vaccinated	0	0	0	0	0	0	411	618	3506
Vaccine Immune	0	0	0	0	0	0	145	141	2571

Source: Study simulations results.

**Table 7 animals-14-00362-t007:** Cost of vaccination in USD.

	Mean	Standard Deviation	Low	High
Vaccination Cost—Setup	141,325	162,207	23,700	1,662,950
Cost of Vaccine	44,677	30,836	22,890	331,645
Total Vaccination Cost	186,002	192,610	46,590	1,994,595

Source: Study simulations results.

## Data Availability

The data presented in this study are available in Appendix A.

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
