# Peer review of "Epidemiology and Scenario Simulations of the Middle East Respiratory Syndrome Corona Virus (MERS-CoV) Disease Spread and Control for Dromedary Camels in United Arab Emirates (UAE)"

_animals, 2024, doi:10.3390/ani14030362_

Round 1
Reviewer 1 Report
Comments and Suggestions for Authors
A well-presented paper. Results under scenarios 2 and 3 are presented well but it is contradicting the real-life scenario. It would be clearer to the reader if the current findings were compared to the published, proven MERS data.
Author Response
Comments and Suggestions for Authors:
A well-presented paper. Results under scenarios 2 and 3 are presented well but it is contradicting the real-life scenario. It would be clearer to the reader if the current findings were compared to the published, proven MERS data.
Response:
Thank you for pointing out this important validation. We have added results from 6 newly added to the manuscript. Recent publications address the real-life spread of MERS-CoV disease and its consequences in the UAE. Please see page 8 of the revised manuscript, lines 34–48, and page 9, lines 1–25. Furthermore, several changes have been made to enhance the manuscript research design, methods description, and the presentation of the results.
Reviewer 2 Report
Comments and Suggestions for Authors
The work of Ali et. al. Titled "Epidemiology and Scenario Simulations of the Middle East Respiratory Syndrome Corona Virus (MERS – CoV) Disease Spread and Control for Dromedary Camels in United Arab Emirates (UAE)" Although it is interesting, it has information that needs to be improved.
· All acronyms must be explained the first time they appear.
· Conclusions are not presented in the abstract. The abstract must be rewritten indicating background, objective, material and methods and conclusions.
· What is the number of cases in each of the periods in Table 2?
· What were the variables used from NAADSM model?
· The Meta-Analysis paragraph (P12, L17-25, table 2) from Yemen and Iraq, are confusing data that do not seem to correspond to this work Scenario Simulators of the Middle East 2 Respiratory Syndrome Corona Virus (MERS – CoV).
· What are the limitations of this work?
· In the conclusions section, the first paragraphs 22-29 do not correspond to conclusions. Conclusions must be rewritten.
Author Response
Comments and Suggestions for Authors
The work of Ali et. al. Titled "Epidemiology and Scenario Simulations of the Middle East Respiratory Syndrome Corona Virus (MERS – CoV) Disease Spread and Control for Dromedary Camels in United Arab Emirates (UAE)" Although it is interesting, it has information that needs to be improved.
Responses:
The authors made several changes to enhance the manuscript background, added several new references, explained the methodology more, and checked the citations for accuracy.
All acronyms must be explained the first time they appear.
Comment addressed. All acronyms are explained the first time they appear.
Conclusions are not presented in the abstract. The abstract must be rewritten indicating background, objective, material and methods, and conclusions.
We agree and have updated the abstract.
What is the number of cases in each of the periods in Table 2?
The cases for each parameter are shown in the column titled “Number of Observation.”
What were the variables used from the NAADSM model?
The authors provided a detailed Excel sheet that includes the variables and parameters used in the NAADSM model in this study as supplementary material to the manuscript.
The Meta-Analysis paragraph (P12, L17-25, table 2) from Yemen and Iraq, are confusing data that do not seem to correspond to this work Scenario Simulators of the Middle East 2 Respiratory Syndrome Corona Virus (MERS – CoV).
From Table 2, only parameters related to UAE are used for modeling the prevalence of MERS-CoV in this paper.
What are the limitations of this work?
The authors included the following paragraph that shows further studies needed, which are limitations of this study.
Further studies needed is to investigate the origin of MER-CoV, not only the host of the virus (the camels). The origins of the virus may include birds and ticks. Studying the ecological connection and environmental conditions that may increase or reduce the spread of the virus. Furthermore, studies are needed to present the importance, scientific methods, trials, and protocols to develop a vaccine against MERS CoV for animals and humans.
In the conclusions section, the first paragraphs 22-29 do not correspond to conclusions. Conclusions must be rewritten.
The authors removed the first paragraph and revised the conclusion.
Reviewer 3 Report
Comments and Suggestions for Authors
1. Why authors do this study?
2. What are the future implications of this study? The abstract should also have a conclusion statement.
3. COVID-19 is also reported in animals, how MERS-CoV has been distinguished from the SARS-CoV2 in current study?
4. What is the test behind the identification of the disease in Camels? Or what kind of symptoms?
5. Is there any statistical analysis used in the present study? Please include the details in the material and methods.
6. Is there any medication provided to the infected camels? Or any vaccine? What kind of vaccine?
7. It seems there are several limitations associated with the current study. Please include a separate section describing the limitations of the study.
Author Response
- Why do authors do this study?
The research objectives of this study are included in section 2 on page 5.
- What are the future implications of this study? The abstract should also have a conclusion statement.
The authors have revised the abstract by adding a conclusion statement
- COVID-19 is also reported in animals, how MERS-CoV has been distinguished from the SARS-CoV2 in current study?
The scope of this study focused on MERS-CoV spread and control strategies within camels’ population in UAE. The study did not include transmission to humans so is not comparable to COVID-19.
- What is the test behind the identification of the disease in Camels? Or what kind of symptoms?
The supplemental Excel sheet added to the manuscript included all parameters assumed concerning the tests for viruses, probability probabilities of observing clinical signs of disease and reporting an observation. Additionally, once a herd is determined to be infected, tracing backward and forward to herds having direct and indirect contact with the infected herd can be performed.
- Is there any statistical analysis used in the present study? Please include the details in the material and methods.
This study is a stochastic simulation of the disease's spread and control strategies using a verified and widely used NAADSM model.
- Is there any medication provided to the infected camels? Or any vaccine? What kind of vaccine?
There is no vaccine developed yet for MERS-CoV.
- It seems there are several limitations associated with the current study. Please include a separate section describing the limitations of the study.
Limitations of the study are included by presenting further studies needed to cover the origin and the ecological and environmental factors that may slow or increase the spread of MER-CoV.
Reviewer 4 Report
Comments and Suggestions for Authors
This reviewer has carefully gone through the manuscript and suggests following points be considered:
1. P2- L1-11: Citations missing.
2. Figure 1: Please mention in the legend if these cases were reported exclusively in the UAE.
3. Same for Figure 2.
4. Figure 3: Numbers (positive confirmed cases) can also be included, to illustrate the disease burden.
5. I think it would be better to compose the research objectives as a paragraph.
6. P6 - L31-33: Please provide a brief context on the life span of camels. And if camels in a particular age group are more prone to MERS infection? Please cite the available literature to support.
7. Figure 6: Please improve the quality of Figure 6; it doesn’t appear to be 300 dpi.
8. Figures 7 and 8: Please provide more information in figure legends.
9. Results: Paragraph 1- please briefly describe the three scenarios before moving forward.
10. Provide the citations for Table 2.
11. Figure 9: Explain the figure in the legend.
12. Explain the sources (citations) of Tables 3 to 7.
13. Conclusions: L22: first discovered or detected?
14. Conclusions must be shortened.
15. In Background, describe the control measures, such as vaccination, that may be in place. Also, briefly describe the MERS disease burden in other Middle Eastern or neighbouring countries and its impact on UAE’s camel populations, in context of trade (live camel trade or meat) or breeding programs etc.
16. The study lacks citations at many places. Authors must pay attention to provide citations when making the claims related to previous studies or data.
17. Abstract can be significantly improved.
18. Acknowledgments, funding, authors contributions etc missing from the manuscript.
Comments on the Quality of English LanguageEnglish can be significantly improved at many places.
Author Response
The authors made several changes to enhance the manuscript background, added several new references, explained the methodology more, and checked the citations for accuracy
Comments and Suggestions for Authors
This reviewer has carefully gone through the manuscript and suggests following points be considered:
- P2- L1-11: Citations missing.
Missing citations have been added.
- Figure 1: Please mention in the legend if these cases were reported exclusively in the UAE.
- Same for Figure 2.
The data displayed in Figures 1 and 2 shows cases reported globally, and the legend is modified accordingly.
- Figure 3: Numbers (positive confirmed cases) can also be included, to illustrate the disease burden.
A new paragraph has been added showing the positive confirmed cases in the Middle East (including UAE) and outside the Middle East, illustrating the disease burden.
- I think it would be better to compose the research objectives as a paragraph.
The research objectives have been composed as a paragraph (section 2).
- P6 - L31-33: Please provide a brief context on the life span of camels. And if camels in a particular age group are more prone to MERS infection? Please cite the available literature to support this.
We have added the suggested content to the manuscript; please see page 6, lines 28-30.
- Figure 6: Please improve the quality of Figure 6; it doesn’t appear to be 300 dpi.
The figures are replaced by one that is higher in resolution.
- Figures 7 and 8: Please provide more information in figure legends.
Descriptions of figures 7 and 8 have been added to the figure’s legends.
- Results: Paragraph 1- please briefly describe the three scenarios before moving forward.
Descriptions of the three scenarios have been added, please see page 9 lines 26-30.
- Provide the citations for Table 2.
Added the source for Table 2 as an unpublished Meta-analysis by the authors in 2022.
- Figure 9: Explain the figure in the legend.
Addressed. Figure 9 legend explained.
- Explain the sources (citations) of Tables 3 to 7.
Sources have been added as “study simulations results”.
- Conclusions: L22: first discovered or detected?
The wording has been corrected.
- Conclusions must be shortened.
The conclusion section has been shortened and updated accordingly.
- In the Background, describe the control measures, such as vaccination, that may be in place. Also, briefly describe the MERS disease burden in other Middle Eastern or neighbouring countries and its impact on UAE’s camel populations, in context of trade (live camel trade or meat) or breeding programs etc.
A new paragraph has been added showing the positive confirmed cases in the Middle East (including UAE) and outside the Middle East, illustrating the disease burden.
- The study lacks citations at many places. Authors must pay attention to provide citations when making the claims related to previous studies or data.
have incorporated your suggestion on citations throughout the manuscript.
- The abstract can be significantly improved.
The abstract has been revised and improved.
- Acknowledgments, funding, authors contributions etc missing from the manuscript.
Acknowledgment, funding, and authors' contributions are now added to the manuscript.
Round 2
Reviewer 2 Report
Comments and Suggestions for Authors
Check the copyright of figures 1 to 6
Author Response
Figures 1 to 6, either publicly published/properly cited or developed by the authors.
Reviewer 3 Report
Comments and Suggestions for Authors
The authors successfully responded to the reviewer's comments and updated the manuscript as well.
Author Response
The manuscript was carefully edited to ensure enhanced readability.
Reviewer 4 Report
Comments and Suggestions for Authors
This reviewer has carefully gone through the revised manuscript. At many places, there is scope of significant improvement in writing. At several places it is still difficult to comprehend. This reviewer suggests that authors carry extensive English editing which will improve the quality of the presentation of work.
I will provide one example which shows the lack of clarity in writing:
1. P9: "humans and camels share a common origin and cross-species transmission ". This is just one example which makes it difficult to comprehend.
2. Author contributions, conflict of interest etc still missing despite authors' claims in the revision letter that these had been added.
Comments on the Quality of English LanguageExtensive editing of English language is required.
Author Response
The manuscript was further edited and corrected for grammar and sentences calrifications. The authors's contributions included.